

# Decoupling climate and avalanche activity: Holocene insights from lacustrine sediments in western Norway

Johannes Hardeng[1], Jostein Bakke[1], Jan M. Cederstrøm[1], Marianne Veste[2]

[1]Department og Earth Science and the Bjerknes Centre for Climate Research, University of Norway, 5007 Bergen, Norway
[2]The Planning and Building Agency, Urban Planning Division, Bergen Municipality, 5020 Bergen, Norway

*Correspondence to*: Johannes Hardeng (johannes.hardeng@uib.no / johannes.hardeng@gmail.com)

**Abstract.** Long-term reconstructions provide critical insights into the interplay between large-scale climatic systems and snow avalanche frequency over millennial timescales. This study presents a ~10,000-year-long snow avalanche record reconstructed from a continuous sediment sequence in a lake in western Norway. A multi-proxy approach, combining bulk sediment analysis (LOI, DBD, and grain size) with high-resolution scanning techniques (MS, XRF, CT scanning), was used to characterise the sediments. The avalanche layers consist of a sand-rich base, dispersed organic fragments, and a fining-
upward sequence of silt and clay. We applied two independent, semi-automatic detection methods to quantify event frequency and sediment influx over time: the rate of change (RoC) of Ti and interactive thresholding of CT greyscale data. Sedimentation in the early Holocene (>10,100 cal yr BP) was dominated by glacial and periglacial processes linked to the study area's deglaciation. A period of low avalanche activity followed during the Holocene Thermal Maximum (>6500 cal yr BP). The avalanche frequency increased during the mid-Holocene (6500-5000 cal yr BP) and intensified further in the late
Holocene, with peak activity from 2300 cal yr BP to the present, a period characterised by large inter-centennial fluctuations and abrupt shifts in avalanche frequency. Comparison with regional palaeoclimatic and palaeo-environmental records suggests that the snow avalanche frequency is primarily controlled by large-scale atmospheric circulation patterns, particularly the intensity of the predominant westerly winds and humidity along the western coast of Norway. These fluctuations are closely linked to North Atlantic Sea surface temperature variability, shifts in the North Atlantic Oscillation,
and broader synoptic climate dynamics.

## 1 Introduction

Snow avalanches are significant natural hazards in alpine and mountainous regions, posing a substantial risk to human life and infrastructure. Mitigation efforts, including forecasting, land-use planning, and structural defences, are crucial to reducing their impact (Øydvin et al., 2011; Taurisano and Øydvin, 2011). However, climate change is challenging the
effectiveness of these measures. The triggering of snow avalanches is governed by the complex interplay between snowpack conditions, terrain, and weather. Key factors include the accumulation and metamorphism of snow, slope steepness, and the presence of weak layers within the snowpack. External triggers, such as intense snowfall, rapid warming, or strong winds, further contribute to instability and the initiation of snow avalanches (Schweizer et al., 2003). These dynamics underscore avalanche activity's sensitivity to short-term weather events and longer-term climatic patterns (Blikra and Selvik, 1998;
Blikra and Sletten, 2002; Nesje et al., 2007).



As the triggering of snow avalanches is inherently weather-dependent, it is closely tied to climatic conditions. Climate change is expected to alter the patterns and magnitude of snow avalanche activity, driven by shifts in temperature, precipitation, and atmospheric dynamics (Hanssen-Bauer et al., 2017). Increased atmospheric instability, including stronger and more frequent winter storms and extreme precipitation events, enhances slab formation and avalanche triggering through rapid snow deposition and wind-driven redistribution. Rising temperatures may shorten the snow accumulation season at lower elevations, as a more significant proportion of the annual precipitation falls as rain rather than snow. At higher elevations, however, increased precipitation is likely to sustain or intensify avalanche activity, potentially with changes in snowpack characteristics that favour a transition towards wetter avalanche types (Castebrunet et al., 2014; Fouinat et al., 2018), such as wet slabs and slush flows (Blikra, 1990). These avalanches differ from their dry-snow counterparts in terms of flow behaviour and may, under certain conditions, exhibit longer runout distances and greater destructive potential. Such developments present new challenges for avalanche forecasting, hazard zoning, and risk mitigation in alpine environments.

Historical events provide valuable insights into these evolving dynamics: During the 1990s, a period dominated by a pronounced positive North Atlantic Oscillation (NAO) phase, several winters were characterised by exceptional snowfall across western Norway (Nesje and Matthews, 2011). In the town of Odda near the study site (Fig. 1), this culminated in large, destructive and fatal wet snow avalanches with exceptionally long runout distances (Gravdal, 1993). These recent events underscore the need to look beyond the instrumental record to better understand how avalanche regimes respond to changing climate conditions (Bradley, 1999).

Snow avalanches erode and entrain material along their tracks, depositing sediments in colluvial fans at the base of slopes. If the runout zone extends into a lake, these sediments may be deposited directly into the basin or temporarily accumulate on lake ice before settling when the ice melts (Sabatier et al., 2022). Once incorporated into the lake sediments, they are often protected from further erosion and preserved in stratigraphic order (Nesje et al., 2007). As such, lacustrine archives enable the reconstruction of long-term variability in snow avalanche frequency. Given the strong climatic control on avalanche formation, these reconstructions are essential for understanding how shifting temperature and precipitation regimes have influenced avalanche dynamics over millennial timescales. In Norway, snow avalanche activity has been investigated through sedimentary sequences preserved in postglacial colluvium (Blikra and Nemec, 1998; Blikra and Selvik, 1998; Aa et al., 2022) and lacustrine sediments (Seierstad et al., 2002; Nesje et al., 2007; Vasskog et al., 2011; Aa et al., 2022). Similar studies from other mountainous regions, such as the Alps, have demonstrated the utility of high-resolution CT scanning for identifying avalanche-derived sediments with increased precision (Fouinat et al., 2017). Reconstructing past avalanche activity in space and time gives valuable insights into the natural range of avalanche variability and contributes to a more robust understanding of future hazard potential under continued climate change.

Here, we present a Holocene-scale reconstruction of snow avalanche activity derived from lacustrine sediments in Lake Vatnasetvatnet in western Norway (Fig. 1A). The region's climate is dominated by its proximity to the North Atlantic, placing it within the pathway of the predominant westerly wind belt (Hurrell et al., 1995). This climatic setting provides an excellent opportunity to investigate the relationship between snow avalanche frequency and the intensity of westerly winds,



humidity, and broader North Atlantic atmospheric dynamics. By integrating sedimentological, geochemical, and physical analysis of the lake sediments, this study aims to provide a more comprehensive understanding of how snow avalanche frequency responds to long-term climatic and environmental changes.

**Figure 1A: The location of the study site in the North Atlantic region, including the major ocean currents influencing the regional**
**climate (elevation data from ESRI Living Atlas). B: The regional map of the Folgefonna Peninsula highlights the study site's position relative to major fjords and the Folgefonna glacier (map data from The Norwegian Mapping Authority). C: Topographic map of the Lake Vatnasetvatnet catchment, an isolated system with limited sediment sources (elevation data from www.hoydedata.no).**



## 2. Study site

Lake Vatnasetvatnet (60.38°N, 6.42°E) is located at 541 m a.s.l. on the western side of the Folgefonna Peninsula in western Norway, north of Folgefonna, the third-largest glacier on mainland Norway (Fig. 1B). The lake covers 0.25 km² and consists of two basins separated by a submerged ridge. The western basin has a depth of 31 m, while the eastern basin extends to 33 m (Fig. 2). The relatively small (1.4 km²) catchment drains westward into the Hardangerfjorden fjord via a small stream. There are no significant river inlets; the water is primarily sourced from local runoff over a gently sloping terrain (<10°) to the west, north, and east, whereas the north face of Vasslifjellet mountain (1056 m a.s.l.) forms the southern catchment boundary. The mountain face has an average slope of 25°, with sections exceeding 50°. The catchment lacks permanent streams, except for two small, intermittent streams that drain gullies on Vasslifjellet and become active only during periods of high runoff. Land cover includes 28% open birch forest, 3% mires, and 51% alpine barren terrain with exposed bedrock, while the lake itself comprises 18% of the catchment (The Norwegian Water and Energy Directorate https://nevina.nve.no/). The bedrock geology of the region is shaped by the Sveconorwegian (~1000 Ma) and Caledonian (~400 Ma) orogenic events, resulting in a complex assemblage of metamorphosed volcanic and sedimentary rocks (Cavalcante et al., 2024). Within the Vatnasetvatnet catchment, the dominant lithologies are metabasalt and quartzite (Ingdal et al., 2001). The landscape reflects extensive glacial erosion and widespread deposition of sediments from the last glacial period (Weichselian). The Scandinavian Ice Sheet reached its maximum extent during the Last Glacial Maximum (LGM, c. 26-19 ka, Clark et al., 2009) and covered even the highest mountain peaks in the area (Regnéll et al., 2021). During the subsequent deglaciation, the ice margin retreated toward the head of Hardangerfjorden before a notable readvance occurred during the Younger Dryas (YD) stadial (Mangerud et al., 2016). This readvance culminated at Halsnøy in outer Hardangerfjorden (Fig. 1B) between 11.8 and 11.6 ka before deglaciation resumed in the early Holocene (Lohne et al., 2012). The ice margin retreated ~120 km through calving, reaching the fjord head within approximately 500 years (Åkesson et al., 2020). Lake Vatnasetvatnet was formed as the ice sheet retreated from the fjord into the surrounding highlands. However, small cirque glaciers persisted in the adjacent mountains until around c. 10 ka (Bakke et al., 2005a; Røthe et al., 2019a).

The climate of western Norway is strongly influenced by the North Atlantic Current (Fig. 1A), which transports warm water from lower latitudes, warming the overlying air masses and increasing atmospheric moisture. This results in a mild maritime climate, with prevailing westerly winds transporting humid air onshore, where orographic uplift leads to high precipitation. The strength of the westerlies and the precipitation patterns are closely linked to variability in the North Atlantic Oscillation (NAO), with a positive NAO phase intensifying westerlies and increasing precipitation, while a negative NAO phase leads to weaker westerlies and drier conditions (Hurrell, 1995; Nesje et al., 2000; Bakke et al., 2008). The Lake Vatnasetvatnet catchment is affected by orographic-induced precipitation, which ranges from 2,500 to 3,100 mm annually, with ~1,000 mm during winter (December to February) and ~500 mm during summer (June to August) (The Norwegian Water and Energy Directorate, https://nevina.nve.no/). The mean annual temperature is 3.7 °C, with February (-3.2 °C) as the coldest month and July (11.5 °C) as the warmest (www.senorge.no).



## 3 Methods

### 3.1 Mapping, lake survey, and coring

The geomorphology of the study area was mapped to identify processes influencing sedimentation in Lake Vatnasetvatnet and to determine potential sediment sources. Avalanche-prone terrain was identified by delineating potential snow accumulation and trigger zones, avalanche tracks, and snow avalanche depositional landforms. Field mapping was conducted during two field seasons, supplemented by LiDAR-derived elevation models (0.25 m resolution; The Norwegian Mapping

Authority, www.hoydedata.no) to generate high-resolution hillshade and slope maps. Changes in vegetation and avalanche track activity were assessed using aerial imagery from different years, dating back to 1971 (www.norgeibilder.no).

The bathymetry and sediment distribution of the lake were surveyed along four west-east and three north-south transects (Fig. 2) using ground-penetrating radar (GPR) equipped with a 25 MHz antenna and GPS. The GPR data were processed using the online version of Mala Vision, employing three filters: DC Offset, background removal, and extensive

amplification with a gain of 300. A velocity of 33 m/ns was applied to calculate water depth and sediment thickness. The GPR data facilitated the selection of coring sites in undisturbed areas with a flat bottom. Three piston cores (VAP109: 294 cm, VAP209: 255 cm, VAP309: 217 cm) were retrieved from the lake ice in 2009 using a modified piston corer (Nesje, 1992) with a 6 m long, 110 mm wide PVC tube. In 2022, an additional sediment core (VAG122: 117 cm) was obtained using a UWITEC gravity corer with a 2 m long, 90 mm wide plastic tube to secure an undisturbed sediment-water interface. Cores

were sectioned, sealed, and stored at 4 °C.

### 3.2 Laboratory analyses

The sediment cores were split, photographed, and visually described to document sediment colour, grain size variations, and lithological units. Physical properties were measured on 1 cm$^3$ bulk samples collected at 0.5 cm intervals from VAP109 and VAP209. Each sample was weighed before and after overnight drying at 105 °C to determine dry bulk density (DBD) and

water content. Loss-on-ignition (LOI) was conducted at 550 °C and reweighed to quantify organic matter content. Based on the DBD and LOI results, 59 and 70 samples from potential event layers in VAP109 and VAP209 were selected for grain size analysis. The samples were treated with hydrogen peroxide ($H_2O_2$) to remove organic material and analysed using a Micrometrics SediGraph 5120. The results were processed using GRADISTAT (Blott and Pye, 2001) in Microsoft Excel.

Magnetic susceptibility (MS) was measured at 2 mm intervals using a Bartington MS2E sensor on a GEOTEK Multi-Sensor

Core Logger (MSCL) (Gunn and Best, 1998) to detect minerogenic input variations. X-ray fluorescence (XRF) scanning was performed to analyse the geochemical composition of the sediments. Cores VAP109, VAP209, and VAG122 were scanned at 30 kV and 50 mA with 1 mm resolution using an ITRAX core scanner (COX Analytics) (Croudace et al., 2006). To account for down-core variations in water content and organic material, elemental count rates were normalised against the total incoherent and coherent scattering (Int + Coh) (Kylander et al., 2011; Davies et al., 2015). Subsequently, the data was

standardised to Z-scores ($Z = (x - \mu) / \sigma$) to ensure consistent comparison across proxies and sediment records. The one mm-



resolution XRF data were smoothed using a 5-point moving average to reduce noise. Computed tomography (CT) scanning was conducted to generate 3D density models of sediment structures in VAP109 and VAG122 using a ProCon XRAY CT-ALPHA. A helical scan was performed with 2,400 images per rotation at 126.5 kV and 800 µA, using a 267 ms exposure time and a 0.5 mm Cu-filter to minimise beam hardening (Brooks and Di Chiro, 1976). The CT projections were

reconstructed using 2-binning to enhance contrast and reduce file size, yielding a final voxel resolution of ~98 µm. A CT greyscale dataset was extracted along a spline through the sediment cores at 0.1 mm resolution, enabling direct comparison with the XRF dataset for event detection.

### 3.3 Chronostratigraphy

For radiocarbon dating, 10 sediment samples were extracted from the sediment cores (*Table 1*). After sieving, terrestrial
macrofossils were identified and dried overnight in sterilised vials before being sent to the Poznań Radiocarbon Laboratory for Accelerator Mass Spectrometry (AMS) analysis. The final age-depth model was constructed using the rBacon package (Blaauw and Christen, 2011) in RStudio v. 4.3.1, employing the IntCal20 northern hemisphere calibration curve (Reimer et al., 2020). Before modelling, event layers were identified following the methodology outlined in Section 3.4. Layers thicker than 3 mm were classified as instantaneous deposits in rBacon. Calibrated ages were estimated at 0.1 cm intervals to match
the resolution of the XRF and CT data.

### 3.4 Event layer identification

We applied two independent approaches to identify event layers in the sediment record. Snow avalanches introduce abrupt pulses of minerogenic material into lakes (e.g., Nesje et al., 2007), resulting in distinct sedimentary signatures that sharply contrast the background sedimentation. To detect these rapid shifts in the sediment composition, we calculated the rate of
change (RoC) for XRF elements by dividing the change in the parameter (δy) by its corresponding time/depth interval (δt) (Støren et al., 2010; Birks, 2012). To further validate these results, we applied interactive thresholding to CT greyscale data, which reflects sediment density, using Thermo Scientific Avizo 2020.2 software (Røthe et al., 2019b; Hardeng et al., 2024). The proportion of sediments exceeding predefined density thresholds was then quantified (volume %) to identify depth intervals corresponding to avalanche layers. Visually distinct minerogenic layers were used to calibrate the RoC threshold
for event detection and the CT greyscale threshold.



# 4 Results

## 4.1 Geomorphology and lake survey

The Vatnasetvatnet catchment (Fig. 2) is a glacially sculpted landscape with thick (>1-1.5 m) till deposits and mires in low-lying areas. The till cover thins with increasing elevation toward Vasslifjellet (1056 m a.s.l.), where large areas of exposed bedrock are present. A well-defined, north-oriented cirque is located 300 m east of Lake Vatnasetvatnet along the catchment's eastern boundary. The boundary between the cirque and the lake catchment is marked by a prominent ridge measuring 8–11 m in height and ~250 m in length. The ridge consists of sub-angular boulders embedded in a matrix of finer-grained sediments. Downslope from the ridge, a channel-like depression approximately 5 m deep extends toward the lake basin.

Figure 2A: Topographic and bathymetric map of the study area, showing coring locations and the two primary sediment sources: Avalanche tracks 1 and 2 (elevation data: www.hoydedata.no). B: Schematic lake map with GPR transects, sediment core locations, and GPR profiles from transects 2 and 7.







**Figure 3: A 3D terrain model of the Vatnasetvatnet catchment (south-facing perspective), showing mapped avalanche tracks (1 and 2) and coring locations. The avalanche tracks are divided into accumulation and trigger zones (red) and colluvial fans in the runout zones (beige). Elevation data from www.hoydedata.no imagery overlay from www.norgeibilder.no.**

Slope processes are concentrated along the southern margin of Lake Vatnasetvatnet, where two well-defined avalanche tracks (Tracks 1 and 2) descend from the northern slopes of Vasslifjellet (Fig. 2). Both tracks originate in north-facing accumulation zones with slope gradients of 30-50° (Fig. 3). From these trigger zones, the tracks extend downslope through treeless corridors where bedrock protrusions and a patchy, thin debris cover indicate ongoing surface erosion (Fig. 4A). The





tracks terminate in runout zones composed of coarse colluvial fans that extend into the lake basin. Trees are absent from the central parts of the colluvial fans, and tree size/age increases progressively towards the outer edges (Fig. 4D). The colluvial fans comprise angular to sub-angular gravel, pebbles, and boulders, with finer sediments perched atop the coarser material (Fig. 4C). We observed snapped trunks, uprooted trees, and patches of sand and gravel atop the vegetation cover along Avalanche Track 1 (Fig. 4B). Both tracks contain small, intermittent streams that drain limited sections of the Vasslifjellet mountain face (0.35 km$^2$ and 0.12 km$^2$). Given their limited drainage area, the contribution of these streams to the influx of allochthonous minerogenic sediments into the lake is minimal.


**Figure 4: Field photographs showing avalanche-related features in the southern part of the Vatnasaetvatnet catchment. A: Overview of Avalanche Tracks 1 and 2 descending from Vasslifjellet into Lake Vatnasetvatnet. B: Broken trees along the edge of Avalanche Track 1 indicate recent avalanche activity. C: The colluvial fan surfaces comprise angular to sub-angular gravel, pebbles, and boulders, often with clasts deposited in unstable positions. D: On the slope, there are distinct, tree-less avalanche corridors. Snow accumulation, likely deposited by a recent small avalanche, is visible below Avalanche Track 2.**





Lake Vatnasetvatnet consists of two distinct basins (maximum depths of 33 and 31 m) separated by a submerged ridge at ~21 m water depth (Fig 2A). The GPR profiles (Fig. 2B) reveal relatively thick (>6 m) sediment infill in both basins, with the highest accumulation in the western basin. The radar stratigraphy displays a continuous sediment sequence over a strong basal reflector, likely representing bedrock or compacted glacial deposits (till). Along the southern lake margin, where the
avalanche tracks terminate, chaotic and hyperbolic reflections indicate coarser, unsorted deposits, likely associated with the colluvial deposits below the avalanche tracks.

## 4.2 Core stratigraphy

Four sediment cores were retrieved from Lake Vatnasetvatnet: VAP109 (294 cm) and VAG122 (117 cm) from the western basin, and VAP209 (255 cm) and VAP309 (217 cm) from the eastern basin (Fig. 2). The sediment record was divided into
two units visually (Fig. 5): Unit A, comprising organic-rich gyttja interspersed with distinct minerogenic layers (event layers 1-25 in the visual sediment log, Fig. 5), and Unit B, which is a finely laminated inorganic silt deposit.

VAP309, retrieved near Avalanche Track 2 in the eastern basin (Fig. 2), contains thick (>10 cm) layers of coarse minerogenic sediments (coarse sand and gravel) with plant fragments. Sharp erosional boundaries indicate sediment remobilisation, likely due to its proximity to the colluvial fan. In contrast, VAP209 cored at a more distal position in the
eastern basin, exhibits stratigraphy comparable to VAP109 from the western basin, with event layers 1-19 correlatable between the two records (Fig. 5). While both VAP109 and VAP209 provide high-resolution and undisturbed sediment sequences, VAP109 is longer and better situated to capture a more balanced record of avalanche deposits, preserving both smaller and larger events. Additionally, geomorphological evidence (Section 4.1) confirms recent avalanche activity in Avalanche Track 1, supporting the focus on the western basin for avalanche reconstruction.

To recover an undisturbed sediment-water interface, gravity core VAG122 was retrieved near the coring location of VAP109 (Fig. 2). The two cores were correlated based on broadly matching sedimentological features, including a sequence of event layers followed by a more homogeneous sediment section without distinct layers. The visual correlation of overlapping sections was not entirely conclusive due to slight discrepancies between the records, which may be attributed to (1) the location of VAG122 closer to Avalanche Track 1, potentially capturing events absent in VAP109, or (2) differential
compaction resulting from varying coring techniques. That said, the correlation is supported by extrapolation of the age-depth model from VAP109, which dates the top of the core to 432 cal yr BP, indicating that approximately 21 cm of sediment is missing. The uppermost section of VAG122 was used to fill the missing interval to account for this; the resulting composite core, VAPG (315 cm), serves as the primary sedimentary record for this study.






Figure 5: Sediment logs and data from VAPG (composite of VAP109 and VAG122) and VAP209. From left to right: core photograph, sedimentological units (A and B), sediment log with visually identified event layers (1-26) used for core correlation, and radiocarbon ages (cal yr BP) at their respective sample depths, with ages in VAP209 transferred from VAPG based on stratigraphic correlation. The graphs show XRF data (Ti, Fe, and Inc/Coh, plotted as Z-scores), MS (SI x 10-5), DBD (g/cm3), water content (wt%), and LOI 550˚C (%).



A Principal Component Analysis (PCA) was conducted to support sediment fingerprinting and aid the classification of samples with similar characteristics into stratigraphic units and facies. Two separate PCAs were conducted, the first for the entire core (Fig. 6A) and the second for just Unit A (Fig. 6B). The analyses included a selection of XRF elemental data (K, Ca, Sr, Ti), the Inc/Coh-ratio, MS, greyscale data, and bulk measurements (LOI, water content, and DBD). Before analysis, the high-resolution scanning data were resampled to match the resolution of the bulk measurements by averaging values

within corresponding 0.5 cm intervals. The first PCA identified two primary clusters (Fig. 6A), which align with the visually defined Units A and B. Samples from Unit B are confined to the lower part of the core (315-260 cm), while Unit A samples correspond to the upper section (above 260 cm, Fig. 5). The first principal component (PC1) accounts for the vast majority of the variance in the dataset (82%). It reflects a gradient in minerogenic content, with strong positive correlations to Sr, Ca, K, Ti, DBD, MS, and greyscale values. Conversely, LOI, water content, and the Inc/Coh ratio negatively correlate with PC1,

as they are associated with more organic-rich sediments. Samples from Unit B are separated from the minerogenic event layers in Unit A by having a generally higher minerogenic content (reflected in higher PC1 scores) and along PC2, which primarily reflects higher MS values.

  The second PCA (Fig. 6B) aimed to separate the visually identified event layers from the gyttja in Unit A. The results show a similar trend as the first PCA, where PC1 account for most of the variability (78%), while PC2 account for 9%. Samples

classified as event layers plot markedly higher along the PC1 axis than gyttja samples, reflecting their minerogenic content. Setting an exact threshold between gyttja and event layer samples is challenging, likely due to the 0.5 cm sampling resolution, which can result in mixed samples. Nevertheless, gyttja is characterised by organic-rich sediment, while the event layers exhibit markedly higher minerogenic input. Ti shows the strongest correlation with PC1 (r = 0.98), thus serving as the most effective proxy for identifying event layers.

Unit B (315-260 cm, Fig. 5) is characterised by finely laminated silt with a high mean dry bulk density (DBD) of 1.56 g/cm$^3$. Loss-on-ignition (LOI) and water content are consistently low, averaging 0.31% and 13.6%, respectively, indicating minimal organic content and a high degree of compaction. MS values are elevated (~94.2 SI x 10$^{-5}$) relative to Unit A. XRF elemental count rates (Z-scores) are generally high (>0) but display substantial variability, reflecting the fine laminations. The XRF count rates have a prominent trend, where the Z-score baseline of Ti decreases from ~3 in the bottom to ~0 near the

transition into Unit A. The incoherence-to-coherence (Inc/Coh) ratio is inversely correlated with the XRF elemental count rates, exhibiting low values (mean: -1.63) with fluctuations corresponding to the laminations. CT greyscale values remain consistently high, indicative of the high density of the sediments. The upper boundary at 260 cm is sharp, marking an abrupt transition into Unit A.

  Unit A (0-260 cm, Fig. 5) consists of dark brown gyttja with relatively high organic content (>30%) and water content

(>70%), resulting in low DBD (<0.30 g/cm$^3$) and MS (mean 21.3 SI x 10$^{-5}$). The LOI is notably higher (>40%) in the lower part of Unit A (260-180 cm), and a weak decreasing trend in water content down-core reflects gradual sediment compaction. XRF elemental count rates generally show lower Z-scores relative to Unit B (<0), while the Inc/Coh ratio remains relatively high (mean: 0.35).





**Figure 6: Principal Component Analysis (PCA) biplots, the proportion of variance explained for each principal component (PC), and every parameter's correlation to PC1 and PC2. A: PCA of the entire VAP118 core, including selected XRF elemental data (K, Ca, Sr, Ti), Inc/Coh ratio, MS, greyscale data, and bulk measurements (LOI, water content, and DBD). Two main clusters correspond to the visually identified Units A and B (Fig. 5). B: PCA, which included only samples from Unit A, was used to evaluate internal variability and distinguish minerogenic event layers from gyttja.**

The minerogenic layers (1-9 cm thick) interspersed within the gyttja in Unit A exhibit distinct physical and geochemical characteristics compared to the surrounding gyttja and represent instantaneous deposits, i.e., event layers (Fig. 7). Their high minerogenic content is reflected by elevated DBD values (0.35 to 0.88 g/cm$^3$, mean 0.48 g/cm$^3$), with correspondingly lower LOI (2.9 to 27.2%, mean 14.2%) and water content (31.3 to 71.7%, mean 62.8%). Several layers contain macrofossils, which is reflected by the high variability in LOI values. MS varies widely between 8 and 121 SI × 10$^{-5}$, with higher values generally associated with the coarser layers. XRF elemental counts show elevated values. Grain size distributions indicate a dominance of coarse silt (25-38%), medium silt (18-30%), and a variable proportion of very coarse silt and sand (>10%),





with overall poor sorting (sorting index ~3.2). The layers consist of a coarse-grained basal layer (sand), in some cases with irregular lower boundaries, indicating potential erosion (Fig. 7). The basal layer grades into a silty deposit with dispersed organic fragments, which is capped by a fining upwards sequence of fine silt and clay.




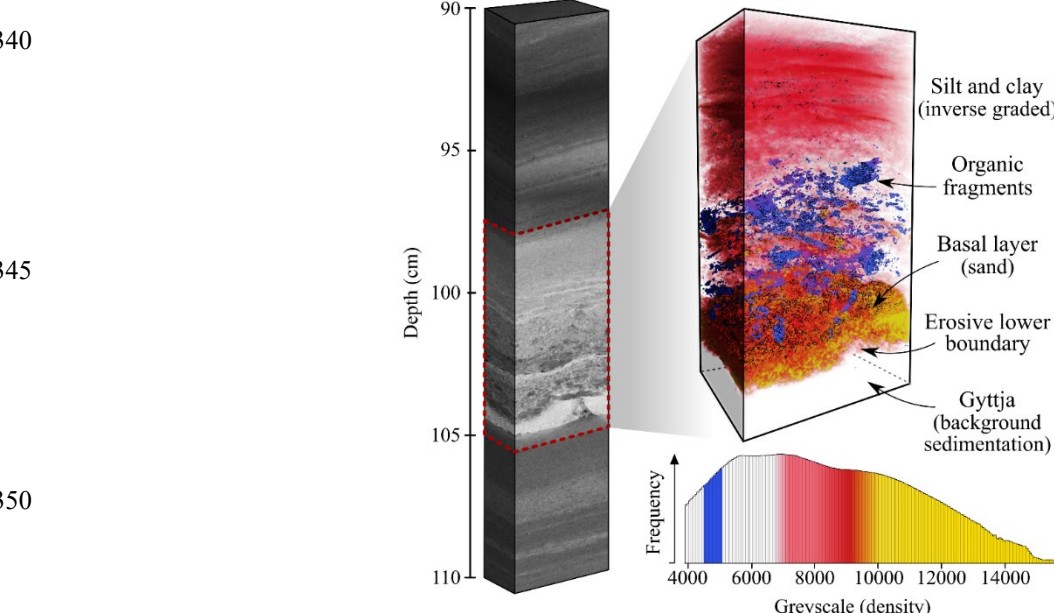

**Figure 7. CT visualisation of an event layer interpreted as snow avalanche deposits. Left: Greyscale CT slice from core VAPG (90-110 cm); light shades indicate higher density. Right: 3D rendering of the internal structure of a snow avalanche layers with organic fragments.**


### 4.3 Chronology and quantifying snow avalanche frequency

Ten macrofossil samples containing leaves, bark, birch fruits and fern leaves were extracted from VAP109 for AMS radiocarbon dating. The resulting radiocarbon ages (Table 1) are in stratigraphic order, with progressively older ages down-core. The oldest sample, obtained at 235.5 cm depth (256.5 cm in the composite VAPG core), has a median calibrated age of

10,109 cal yr BP.

Before age-depth modelling, event layers were identified using the two approaches outlined in section 3.4: the rate of change (RoC) of XRF elemental count rates and interactive thresholding of CT greyscale data (Fig. 8). RoC was calculated for Ti where the threshold for event detection was set to the 95th percentile (rationale in section 5.2). For CT thresholding, we classified depths with more than 40% minerogenic sediment as event layers. Layers >3 mm thick were marked as

instantaneous deposits (*slumps*) in the rBacon age-depth model to improve chronological accuracy and better reflect true depositional ages (Fig. 8). Following the establishment of the age-depth model, the data were resampled to their corresponding age intervals, and the RoC of Ti was recalculated using the change in age as δt. The results of the CT greyscale thresholding were also plotted on the age scale to reconstruct the fraction (volume %) of snow avalanche detritus over time.





The results from the two approaches were summed into 100-, 200-, and 500-year intervals to assess long-term variability in event frequency and influx of snow avalanche detritus (Fig. 9). The two reconstructions show a strong correlation (r = 0.76) for the 100-year summed data, indicating consistency between the methods.

A total of 187 event layers were identified using the RoC approach, and the reconstructed avalanche frequency (Fig. 9) reveals significant variations throughout the Holocene. The early to early-mid Holocene (>6500 cal yr BP) was characterised
by a low frequency, with 0-1 events per 500 years. A marked increase occurred between 6500 and 5100 cal yr BP, reaching 11 events per 500 years, followed by a decline between 5000 and 4200 cal yr BP (1-2 events per 500 years). From 4200 to 2300 cal yr BP, avalanche frequency gradually increased from 4 to 21 events per 500 years. The highest activity occurred in the late Holocene (<2300 cal yr BP), with 23–37 events per 500 years. This period exhibits strong fluctuations, with event frequencies ranging from 1 to 16 events per 100 years. The sedimentation rate (Fig. 8C) varied between 7 and 29 cm per 500
years, mirroring the reconstructed event frequency and the influx of minerogenic sediments. Periods of low sedimentation rates coincide with low event frequencies, while peak sedimentation rates correspond to peak event frequency.

Table 1: Overview of radiocarbon dates from VAP109. The corresponding depths in the composite core (VAPG) are +21 cm.

| Core | Depth (cm) | Lab ID | $^{14}C$ age (BP) | | Min. age (-1σ) (cal BP) | Max. age (+1σ) (cal BP) | Median probability (cal BP) | Sample material |
|---|---|---|---|---|---|---|---|---|
| VAP109 | 9.5 | Poz-33998 | 575 35 | +/- | 527 | 646 | 600 | birch fruits and leaves |
| VAP109 | 26.5 | Poz-33999 | 1110 40 | +/- | 928 | 1175 | 1013 | fern leaves, birch fruits |
| VAP109 | 59.5 | Poz-34001 | 1830 30 | +/- | 1631 | 1823 | 1732 | birch fruits and leaves, dwarf birch |
| VAP109 | 69.5 | Poz-34002 | 1905 30 | +/- | 1732 | 1920 | 1809 | birch fruits and leaves, bark |
| VAP109 | 87.5 | Poz-34003 | 2195 35 | +/- | 2112 | 2329 | 2226 | birch fruits and leaves, bark |
| VAP109 | 134.5 | Poz-34004 | 3735 35 | +/- | 3978 | 4230 | 4087 | birch fruits and leaves, bark |



| | | | | | | | | |
|---|---|---|---|---|---|---|---|---|
| VAP109 | 158.5 | Poz-34005 | 4880 +/- 40 | 5483 | 5719 | 5617 | | birch fruits and leaves, small twig |
| VAP109 | 189.5 | Poz-34006 | 5640 +/- 40 | 6310 | 6494 | 6419 | | fern leaves, birch leaves |
| VAP109 | 231.5 | Poz-34007 | 8400 +/- 50 | 9295 | 9528 | 9432 | | birch fruits and leaves, bark |
| VAP109 | 235.5 | Poz-34008 | 8750 +/- 50 | 9548 | 10109 | 9736 | | birch leaves |

**Figure 8: Identification of event layers in VAPG using interactive thresholding of CT greyscale data and the 95th percentile for Ti's rate of change (RoC). Event layers >3 mm (shaded grey bars) were specified as instantaneous deposits in the age-depth model.**





**Figure 9: Holocene avalanche reconstruction from Lake Vatnasetvatnet. A: Number of identified avalanche layers over time, summed using 100-year, 200-year, and 500-year bins. The lower panel shows the rate of change (RoC) in Ti content with the 95th percentile, used to detect abrupt sedimentological shifts, i.e., avalanche layers. B: Avalanche-derived sediment deposition based on thresholded CT data. Note that the sediment volume is the summed volume % (lower panel) for 100, 200, and 500 years and is a relative scale with no units. (C) Sedimentation rate (cm per 500 years).**

## 5 Discussion

### 5.1 Catchment processes

Lake Vatnasetvatnet is well-suited for recording episodic slope processes due to its isolated setting and lack of significant fluvial inflow. Steep terrain is confined to the south of the lake, making it relatively straightforward to distinguish potential




sediment sources. Postglacial landscape evolution has been limited, with only small, intermittent streams and no evidence of substantial fluvial reworking.

The small cirque located approximately 300 m to the east (Fig. 2) likely hosted a local cirque glacier during the final stages

of deglaciation in the early Holocene. Its north-facing orientation is favourable for preserving glacial ice by promoting snow accumulation from the prevailing southwesterly winds and reducing solar radiation, similar to the sites described in Larsen et al. (1984). The morphology, composition, and position relative to the cirque suggest that the observed ridge represents a lateral moraine. The shallow depression extending downslope from the ridge likely functioned as a meltwater outlet feeding into the lake basin.

Like the cirque, the two well-defined avalanche tracks (Figs. 2-4), with associated accumulation and trigger zones, have a north-facing aspect – sheltered from the predominant southwesterly winter storms, creating optimal conditions for slab formation through wind redistribution of snow. Slope gradients range from 30° to 50° (Fig. 3), which is gentle enough to permit snow accumulation yet steep enough for avalanches to trigger (Martinelli, 1974; Luckman, 1977). The tracks are treeless corridors with visible signs of active surface erosion, indicative of frequent avalanche activity. The absence of trees

in the central parts of the runout zones, coupled with progressively older trees toward the margins, suggests regular, channelised avalanches punctuated by less frequent high-magnitude events with a broader impact. Additional evidence of recent activity includes snapped trees and gravel deposits perched atop the vegetation. The colluvial fans exhibit characteristic avalanche-derived sedimentology with poorly sorted material, including sub-angular and angular gravel, pebbles, and boulders. Gravel and pebbles are deposited in unstable positions in many places due to the melting of sediment-

laden avalanche snow (Luckman, 1978; Blikra and Nemec, 1998). No morphological indicators of debris flow activity were observed, such as lobes or levees (e.g., Sharp, 1942; Blikra and Nemec, 1998). Based on these combined geomorphological and sedimentological observations, we argue that snow avalanches have been the predominant source of allochthonous minerogenic sediment to Lake Vatnasetvatnet since deglaciation.

**5.2 Interpretation of the sediment record**

The finely laminated silt in the lower part of the core (Unit B, Fig. 5) is interpreted as glaciolacustrine sediments deposited during the deglaciation of the Vatnasetvatnet catchment. The consistently high DBD values (>1.5 g/cm$^3$) and low organic content, as indicated by LOI (~0.3%), are characteristic of glaciolacustrine sedimentation (e.g., Karlén, 1976; Nesje et al., 1991; Bakke et al., 2005b). Additionally, Unit B exhibits high XRF elemental count rates, elevated MS values, and peak CT greyscale values, all of which are indicative of the dense, minerogenic sediments associated with glacial environments (Fig.

6). Fluctuations in the XRF parameters and MS are the primary drivers for the samples' position along the PC2 axis (Fig. 6). This variability is prominent in the upper half of Unit B and likely reflect pulses of sediment supply, potentially related to seasonal meltwater influx and varve formation (e.g., Regnéll et al., 2019). The trend in decreasing XRF count rates towards the transition into Unit A reflects a decrease in the mean grain size, possibly indicating a gradually more stagnant glaciolacustrine setting as the glacier retreated. The abrupt boundary between Units A and B represents the end of glacial



influence, marking the transition from a glaciolacustrine environment into a lacustrine environment. The last influence of glacial processes was likely the small glacier inhabiting the cirque directly east of the catchment. The minimum age of this final deglaciation is constrained to ~10,100 cal yr BP, which is supported by previous studies from the Folgefonna glacier (Bakke et al., 2005a; Røthe et al., 2019a).

Unit A is characterised by organic-rich gyttja with clay, which we interpret as the background sedimentation. The high LOI values indicate that the in situ production of organic material dominates the background sedimentation. Higher LOI values in the lower part of Unit A suggest that the in situ organic production rate was higher during the early Holocene, likely related to the elevated summer temperatures during the Holocene Thermal Maximum (HTM) (e.g., Bjune et al., 2005; Seppä et al., 2009). In contrast, the event layers have distinct physical and geochemical signatures that reflect their high-energy depositional origin. The PCA (Fig. 6) separates these layers from the background sedimentation along PC1, which correlates strongly with variables indicative of minerogenic input, most notably Ti (r = 0.98).

We interpret the predominant depositional process of these event layers as wet snow avalanches entering an ice-free lake, most likely during the late winter or spring melt episodes. Wet snow avalanches are dense, water-saturated flows. They are particularly effective erosional agents, capable of mobilising a wide range of material, from fine particles to boulders and terrestrial organic matter (Rapp, 1960). The layers are poorly sorted, with grain sizes ranging from sand to clay, and commonly exhibit a structured sequence: a basal, sand-rich unit with an irregular lower boundary. This basal unit grades into silt-rich material with dispersed macrofossils (e.g., twigs, grass, leaves), which is capped by a fining-upward sequence of silt and clay (Fig. 7). The presence of terrestrial organic material and elevated LOI values (up to 27.2%) in some layers indicates that these events entrained vegetation during downslope transport. The observed stratigraphy reflects rapid settling from a turbulent sediment cloud, with coarse clasts and organic fragments deposited first, followed by finer suspended particles. While the sharp basal boundary may indicate erosion, the age-depth model does not indicate significant hiatuses, suggesting that these contacts reflect compaction or settling dynamics rather than reworking. Their irregular basal contacts argue against deposition as drop-stone from melting lake ice (Luckman, 1975; Nesje et al., 2007). Instead, these features point to sudden, high-momentum sediment delivery by water-saturated snowflows that entered the lake directly or, potentially, by breaching lake ice upon impact (Sabatier et al., 2022).

### 5.3 Quantifying snow avalanche activity

Quantifying snow avalanche activity from lacustrine sediment records requires a robust methodological approach to distinguish episodic event layers from background sedimentation. In this study, we applied two independent detection methods: (1) rate of change (RoC) analysis of Ti as a proxy for abrupt minerogenic influx and (2) adaptive thresholding of CT greyscale values to identify and quantify density-based variations in sediment composition (Fig. 8). Snow avalanches introduce sudden pulses of minerogenic material into the lake basin, significantly altering the depositional conditions (Sabatier et al., 2022). Our detection methods are based on the premise that event layers are abruptly deposited, exhibit a





distinct density contrast relative to the background sedimentation, and show strong co-variability with peak values in both XRF elemental counts and CT greyscale data (Figs. 5 and 6).

Rate of change (RoC) analysis has proven to be an effective tool for detecting the abrupt sedimentary shifts associated with
episodic events (e.g., Støren et al., 2010; Røthe et al., 2019b; Johansson et al., 2020; Hardeng et al., 2022). Applying consistent criteria for defining avalanche layers eliminates the need for subjective assessment of individual layers throughout the sediment record. Selecting the RoC threshold requires careful calibration to avoid under- or overestimating event frequency. For this study, we computed the RoC of Ti, a parameter well suited for distinguishing minerogenic layers from gyttja due to its low signal-to-noise ratio and the sharp contrast between baseline (background) and peak values (Fig. 5 and
6). Multiple RoC thresholds were tested against the sediment log and CT images. The 95$^{th}$ percentile was selected as the optimal threshold, capturing all visually identified minerogenic layers while minimising noise effects.

CT scanning is a well-suited method to identify the stark density difference between the inorganic, dense avalanche layers and the low-density organic gyttja. Although initially developed for medical applications, CT scanning has become a valuable tool for identifying and quantifying deposits and structures in sediment cores, including tephra (van der Bilt et al.,
2021), ice-rafted debris (Cederstrøm et al., 2021), varves (Ballo et al., 2023), floods (Støren et al., 2010; Hardeng et al., 2022; 2024), and snow avalanches (Fouinat et al., 2017; Røthe et al., 2019b). Similar to setting the RoC threshold for distinguishing background sedimentation from event layers, defining the CT greyscale threshold for minerogenic sediment involves a degree of subjectivity. Additionally, CT greyscale values are not directly comparable between core sections, as each scan generates its unique greyscale distribution (van der Bilt et al., 2021). As such, a separate threshold calibration for
each core section is required. To aid the process, we used the visual core descriptions and bulk samples to identify depths containing minerogenic sediments, ensuring consistent and reliable threshold calibrations.

Despite the inherent subjectivity in defining threshold values, both methods allow for objective and reproducible event detection once the threshold is established. The RoC approach quantifies the frequency of discrete snow avalanche layers, while CT thresholding provides the relative proportion of minerogenic sediment input over time. The strong correlation (r =
0.76) between these independent reconstructions supports the robustness of our methodology and reinforces the validity of a multi-proxy approach for quantifying snow avalanche frequency.

**5.4 Palaeoclimatic implications**

The snow avalanche reconstruction from Lake Vatnasetvatnet (Fig. 9) reveals significant centennial to millennial-scale variability throughout the Holocene. This variability exhibits strong co-variability with regional glacier dynamics, flood
frequencies in river systems in southern Norway, and broader North Atlantic climate trends (Fig. 10), highlighting the interconnected nature of these systems. However, caution is warranted when interpreting avalanche frequency solely as a proxy for climatic conditions. The lake's sensitivity to external forcing is modulated by local factors such as vegetation cover, topography, and sediment availability (Rubensdotter and Rosqvist, 2009). In particular, the position of the local tree line may influence avalanche release, as vegetation can stabilize the snowpack and reduce avalanche frequency. Shifts in





vegetation may result from natural climate variability but can also reflect anthropogenic land-use changes, such as grazing or
deforestation, which may alter slope stability and snow dynamics (Rapuc et al., 2024).

### 5.4.1 Early to early-mid Holocene (>6500 cal yr BP): Warm summers, cold and dry winters

The early to early-mid Holocene (>6500 cal yr BP) was characterised by low snow avalanche frequency, coinciding with the
Holocene Thermal Maximum (HTM). This period was marked by high summer insolation (Laskar et al., 2004, Fig. 10A),
elevated summer sea-surface temperatures in the North Atlantic (e.g., Calvo et al., 2002), and increasing terrestrial
temperatures in western Norway (Bjune et al., 2005) and northern Europe (Seppä et al., 2009). However, winter insolation
was lower than present, as reflected in terrestrial winter temperature reconstructions (Davis et al., 2003, Fig. 10C). Similarly,
subsurface temperature reconstructions from the Norwegian Sea (Dolven et al., 2002; Risebrobakken et al., 2003), argued to
reflect annual mean or winter temperatures (Jansen et al., 2008; Andersson et al., 2010; Risebrobakken et al., 2011) were
colder than present (Fig. 10B).

During this period, glaciers in Norway largely disappeared due to a combination of high summer temperatures and low
winter precipitation (Nesje et al., 2008a). This is also reflected by the equilibrium line altitude (ELA) reconstruction from
Folgefonna (Bakke et al., 2005a; Røthe et al., 2019a). The low frequency of snow avalanche deposits at Lake Vatnasetvatnet
(Fig. 10H) mirrors this broader regional pattern, with similarly low avalanche activity recorded in other lacustrine
reconstructions from western Norway (Nesje et al., 2007; Vasskog et al., 2011, Fig. 10E). Additionally, river systems
predominantly controlled by snowmelt floods also recorded low flood activity during this period (e.g., Bøe et al., 2006;
Støren et al., 2010; 2016; Paasche and Støren, 2014; Hardeng et al., 2022; 2024, Fig. 10F-G). The low snow avalanche
activity during the HTM may partly reflect a higher tree line (Bjune et al., 2005), which could have contributed to snowpack
stabilization; however, while this effect cannot be entirely ruled out, it is unlikely to fully account for the prolonged
reduction in activity observed in the record.

### 5.4.2 Middle Holocene (6500–4200 cal yr BP): Warmer and more humid winters

Between 6500 and 4200 cal yr BP, winter temperatures increased relative to the early Holocene (Davis et al., 2003),
accompanied by a marked rise in avalanche frequency. The number of detected events increased from 0-1 per 500 years to
11 per 500 years around 5500 yr BP. This period coincides with increased glacial activity at Folgefonna (Bakke et al., 2005a;
Røthe et al., 2019a), with a lag of 200-300 years, suggesting that the glacier advanced after a period of increased winter
precipitation. The observed increase in avalanche activity aligns with other avalanche reconstructions in western Norway
(Vasskog et al., 2011) and overlaps with elevated flood frequency in the predominantly snowmelt-driven flood regimes in
southeastern Norway (Støren et al., 2010; Hardeng et al., 2024). A decline in avalanche activity between 5300 and 4200 cal
yr BP (to 1-2 events per 500 years) is followed by a temporary reduction in glacial activity at Folgefonna, suggesting a brief
period of decreased winter precipitation or a shift in storm trajectories.



### 5.4.3 Late Holocene (4200 cal yr BP-present): Increasing winter temperature and precipitation

From 4200 cal yr BP, snow avalanche frequency exhibits a steady increase. The highest recorded activity occurred from 2000 cal yr BP to the present. This period corresponds with a transition from high summer insolation and low winter insolation to increased winter insolation and reduced summer insolation (Laskar et al., 2004). The long-term trend of increasing avalanche frequency is also observed in the other avalanche reconstructions in western Norway (Vasskog et al., 2011) and snowmelt-driven flooding in southern Norway (Bøe et al., 2006; Støren et al., 2010; Engeland et al., 2020; Hardeng et al., 2022; 2024), reinforcing the role of enhanced winter precipitation (Paasche and Støren, 2014; Støren and Paasche, 2014; Støren et al., 2016).

Glacial expansion across western Norway, including at Folgefonna, further suggests that this period was characterised by colder summers and wetter winters. Over the past 1500 years, the relationship between avalanche frequency and winter temperature has been more pronounced, with peak winter temperatures recorded in the Norwegian Sea during the Little Ice Age (LIA; 18th-19th centuries) (Fig. 10B). Historical records from this period document higher frequency of hazardous events, including snow avalanches, floods, and rapid glacial advances (Grove, 1972; 1983; Nesje et al., 2008b). Although the increased frequency of snow avalanche deposits during the LIA likely reflects warmer winters and enhanced precipitation, grazing from over the past ~300 years may have reduced vegetation cover and increased slope erosion, potentially amplifying sediment delivery and contributing to an apparent increase in event frequency (Rapuc et al., 2024). However, given the extensive historical evidence of increased snow avalanche activity during this period, it is unlikely that land-use changes alone account for the increased influx of minerogenic sediments.

### 5.4.4 Hydrological intensification and the role of the North Atlantic Oscillation (NAO)

The observed co-variability between snow avalanche activity, snowmelt-induced floods, and glacier dynamics suggests that the same climatic drivers influence them. Winter precipitation is the common denominator between snowmelt floods, glacier mass balance, and snow avalanches. High winter precipitation, coupled with frequent storm activity, promotes avalanche formation by increasing the accumulation of unstable snowpacks in steep, high-relief terrain. The long-term increase in avalanche frequency corresponds with increasing winter temperatures, suggesting that hydrological intensification, primarily driven by gradually increasing insolation, plays a significant role in millennial-timescale shifts in avalanche activity. Warmer ocean conditions in the North Atlantic (Fig. 10B) enhance evaporation and atmospheric moisture content (Trenberth et al., 2003; Willett et al., 2008; O'Gorman and Schneider, 2009), increasing winter precipitation over western Norway.

On shorter (centennial) timescales, the avalanche variability is likely modulated by the prevailing atmospheric circulation patterns, particularly the NAO. The NAO strongly influences winter precipitation patterns in Norway, with a positive NAO phase leading to intensified westerlies and increased precipitation, while a negative NAO phase weakens the westerlies, reducing precipitation (Hurrell, 1995; Hanssen-Bauer and Førland, 2000; Goslin et al., 2018). During negative NAO phases, synoptic systems track further south, delivering enhanced precipitation to western Norway while reducing moisture transport



to central and eastern Europe (Bakke et al., 2008; Støren et al., 2012). A comparison of NAO reconstructions (e.g., Olsen et al., 2012) with avalanche frequency trends suggests a strong link, particularly during the late Holocene when negative NAO
conditions coincided with increased winter storminess and snowfall.

### 5.4.5 Implications for future snow avalanche activity

The Holocene avalanche reconstruction provides a valuable analogue for assessing potential changes in future avalanche frequency. Climate change alters the seasonality and magnitude of extreme winter precipitation, with important implications for snow avalanche activity (Hanssen-Bauer et al., 2015; 2017; Fouinat et al., 2018). While rising temperatures will reduce
overall snow cover at lower elevations, they may contribute to increased snow accumulation and instability at higher elevations (Vormoor et al., 2015), intensifying avalanche activity in some regions. Future warming may also alter the snow avalanche dynamics, moving from more dry snow avalanches to more rainfall-triggered wet avalanches (Ballesteros-Cánovas et al., 2018; Mayer et al., 2024). Wet snow has a high density relative to dry slabs, and the presence of water reduces friction (Salm, 1982), which may lead to increased runout distances – a crucial consideration in land-use planning
and snow avalanche mitigation.

A key uncertainty is the future trajectory of the NAO and storminess. If climate change leads to a more persistent positive NAO, winter precipitation may continue to increase. Delworth and Zeng (2016) showed that prolonged negative NAO periods can strengthen the Atlantic Meridional Overturning Circulation (AMOC). This will intensify the northward ocean heat transport via the North Atlantic current (Fig. 1A), which may lead to higher evaporation rates and increased atmospheric
moisture content. However, if warming disrupts current atmospheric circulation patterns, the response of avalanche activity may become more complex and uncertain. The findings in this study underscore the importance of considering both temperature and precipitation changes when assessing future avalanche hazards.

While declining snow cover may reduce avalanche frequency in some areas, enhanced winter precipitation and hydrological intensification could sustain avalanche activity in high-relief coastal regions such as western Norway. Continued monitoring
and modelling efforts are crucial for refining our understanding of snow avalanche-climate interactions.





**Figure 10: Comparison between the snow avalanche reconstruction from Lake Vatnasetvatnet (this study) and regional palaeoclimatic and palaeoenvironmental records. A: Winter (ONDJFM) and summer (AMJJAS) insolation at 65°N (Laskar et al., 2004). B: Sub-surface temperature reconstruction from the Norwegian Sea (Dolven et al., 2002). C: Pollen-based annual mean temperature anomaly for north-western Europe (Davis et al., 2003). D: Equilibrium line altitude (ELA) reconstruction from Folgefonna glacier, southwestern Norway (Bakke et al., 2005a; Røthe et al., 2019a). E: Snow avalanche reconstructions showing the number of snow avalanche particles/100 years from Lake Vanndalsvatnet, western Norway (Nesje et al., 2007) (grey bars) and the number of snow avalanche layers/100 years from Lake Oldevatnet, western Norway (Vasskog et al., 2011) (white bars). F: Flood frequency reconstruction from Lake Berse, southeastern Norway (Hardeng et al., 2024). G: The predominant flooding regime reconstruction from Lake Lygne, southwestern Norway, expressed as the difference between the number of snowmelt and rainfall floods (Hardeng et al., 2022). H: Snow avalanche frequency reconstruction from this study, using the rate of change of Ti as a proxy for snow avalanche layers.**

## 6 Conclusions

- By combining bulk sediment analysis (LOI, DBD, and grain size) with high-resolution scanning techniques (MS, XRF, CT scanning), we have robustly classified the sediment record from Lake Vatnasetvatnet. Sedimentation in the early Holocene (>10,100 cal yr BP) was dominated by glacial and periglacial processes linked to the deglaciation of the study area. Since then, the primary source of allochthonous sediments in this enclosed system has been snow avalanches triggered from the north face of the Vasslifjellet mountain. The avalanche layers are dense, minerogenic deposits dominated by coarse silt and fine sand, exhibiting peak XRF elemental count rates and CT greyscale values.

- To quantify avalanche frequency and sediment influx over time, we applied two independent, semi-automatic detection methods: the rate of change (RoC) of Ti and interactive thresholding of CT greyscale data. There is a strong positive correlation (r = 0.76) between these independent reconstructions, which supports the robustness of our methodology.

- The snow avalanche activity at Vasslifjellet reflects regional climatic variability, displaying strong co-variability with glacial activity, flood frequency, and other palaeoclimatic records from western Norway and the broader North Atlantic region.

- Early to early-mid Holocene (>6500 cal yr BP) was characterised by warm summers and cold, arid winters. During this period, snow avalanche frequency remained low (0-1 events per 500 years), coinciding with glacier retreat across western Norway and a period of low flood frequency.

- Mid-Holocene (6500-4200 cal yr BP) had increasing winter temperatures and precipitation. This led to a rise in snow avalanche frequency, reaching 11 events per 500 years before declining between 5000-4200 cal yr BP, a pattern mirrored in other records sensitive to winter precipitation.

- Late Holocene (4200 cal yr BP-present) saw a shift toward wetter and milder winters, leading to a steady increase in avalanche activity. From 4200 to 2300 cal yr BP, avalanche frequency gradually increased from 4 to 21 events per 500 years. The highest activity occurred in the late Holocene (<2300 cal yr BP), with 23–37 events per 500 years.

- The long-term (multi-millennial) variability in avalanche frequency is likely linked to variations in winter precipitation, driven by hydrological intensification associated with solar forcing and rising winter temperatures. On shorter time scales (centennial), the activity is modulated by large-scale atmospheric circulation patterns, particularly the North

Atlantic Oscillation (NAO). Understanding the climatic controls behind past avalanche activity provides key insights into long-term winter hydroclimate variability.

- Climate warming is expected to increase extreme winter precipitation events, potentially sustaining or intensifying
avalanche activity at higher elevations while shifting avalanche dynamics toward more rainfall-triggered wet avalanches. These findings highlight the need for continued monitoring and updated hazard assessments to account for evolving snow avalanche risks in a changing climate.

**Data availability**

The raw data generated and used in this study will be submitted to the PANGAEA database (https://pangaea.de/).

**Author contributions**

JH, MV, and JB designed the conceptual framework for the study. JB and MV conducted the initial coring and field surveys in 2009, while JH carried out additional field surveys in September 2022. The coring in March 2022 was led by JH, with field assistance from JMC. MV conducted extensive laboratory work, and JMC conducted the CT scanning and interactive thresholding. JH performed the data analysis, compiled figures, and led the writing of the manuscript. All authors contributed to the scientific discussions, data interpretation, and manuscript revisions and have approved the submitted
version.

**Declaration of competing interest**

The authors declare that they have no known competing financial interests or personal relationships that could have influenced the work reported in this paper.

**Acknowledgements**

We thank Bjørn Christian Kvisvik for participating in the coring campaign in 2009. We also want to thank Ingelinn Aarnes for helping us identify the terrestrial macrofossils used for radiocarbon dating. All laboratory analyses, except for dating, were done at the National Infrastructure EARTHLAB (NRC 226171) at the University of Bergen. Artificial intelligence assistance: During the preparation of this manuscript, language and editing support was provided by ChatGPT (OpenAI), including assistance with phrasing, clarity, and consistency of tone. All scientific content, interpretation, and conclusions
remain solely the product and responsibility of the authors.





**Funding**

This study was supported by the National Infrastructure EARTHLAB (NRC 226171) at the University of Bergen.

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
