# Peer review of "Decoupling climate and avalanche activity: Holocene insights from lacustrine sediments in western Norway"

_EGUsphere, 2025_

## Author Comment (AC2)

**Anonymous Referee #1:**

(Referee comments in regular font, author response in italic gray font)

This well written and interesting paper investigates the sedimentary record of a high-mountain lake in Western Norway. The author focus on the event-layer chronology and put this in the context of avalanche frequency as a result of climate and weather patterns. They identify three intriguing periods in the Holocene when avalanche activity was highly different. These findings are coinciding with patterns of flood and glacial activity. Moreover, there seems to be a clear link to NAO on a centennial time scale, with all of these players influencing the moisture distribution, temperature (snowline!) and atmospheric circulation patterns. The topic is highly relevant for the Journal Climate of the Past and I was pleased to read this well-structured and well written contribution.

We thank the reviewer for their positive and encouraging assessment of the manuscript. We appreciate the recognition of the study's clarity, structure, and relevance, and are pleased that the broader climatic context and NAO link were well received.

I do have some key comments upfront, followed by some detailed comments.

I welcome the applied dual method of identifying event layers (CT thresholding, RoC) minimizing subjectivity. However, my main concern is that the authors interpret all event layers solely as avalanche-induced ("wet snow avalanches entering an ice-free lake"), and do not consider them being deposited by extreme floods caused by precipitation events. There is no doubt that this basin is influenced by avalanches, which leave some traces in the sedimentary record. However, I am missing a discussion on extreme precipitation events, and how the respective layer can be distinguished. The authors mention that "only small, intermittent streams and no evidence of substantial fluvial reworking" occur and "Given their limited drainage area, the contribution of these streams to the influx of allochthonous minerogenic sediments into the lake is minimal". But there are plenty of high-mountain lake-sediment records that are clearly punctuated by flood deposits, in particular also in basins with no permanent inflow. These lakes also lack major deltas or alluvial fans, as is the case of the Lake Vatnasetvatnet, but this does not mean that extreme precipitation events do not deliver minerogenic deposits to the basin.

We thank the reviewer for raising this critical point and discussing possible alternative interpretations. We fully acknowledge that distinguishing avalanche-derived deposits from flood-related turbidites can be challenging in lacustrine archives, particularly when both processes may produce fining-upward, minerogenic layers. However, we remain confident that the event layers in Lake Vatnasetvatnet reflect avalanche activity. The rationale for this interpretation is outlined in Sections 5.1 Catchment processes and 5.2 Interpretation of the sediment record. To clarify, we do not state that snow avalanches are the only source of allochthonous minerogenic material, but we argue that they are the predominant source, as stated in lines 441–443: "Based on these combined geomorphological and sedimentological observations, we argue that snow avalanches have been the predominant source of allochthonous minerogenic sediment to Lake Vatnasetvatnet since deglaciation."

We recognise that extreme precipitation events can temporarily activate the intermittent streams and mobilise small amounts of sediment, but such events are unlikely to generate the magnitude of sediment influx and energy required to form the distinct event layers and grain sizes observed at the coring sites. There are no traces of such erosive events in the catchment.

While the reviewer refers to other high-mountain basins where floods have produced comparable deposits, no specific studies are cited, and we are not aware of any studies with directly comparable settings. The most relevant analogue is the study by Nesje et al. (2007, The Holocene), which reconstructed snow-avalanche activity from a similarly small and isolated basin in western Norway. Notably, that study did not identify any flood-related deposits, further supporting our interpretation that such processes are unlikely in the Vatnasetvatnet catchment. The slopes surrounding Vatnasetvatnet are covered by thin compact glacial till, stabilised by a vegetation cover, and offer little erodible material for running water. Limited erosion may occur along the intermittent streams, and vegetated avalanche fans themselves during the most intense rainstorms, yet these areas represent only a minor sediment source. The geomorphic context at around Lake Vatnasetvatnet, characterised by thick vegetation, compact till, and scarce sediment along the stream channels, renders such flood-induced deposits improbable.

The event layers contain terrestrial macrofossils, which we interpret as material entrained by snow avalanches traversing the vegetated slopes. By contrast, rainfall in this setting would not inundate broad areas; water would percolate through vegetation and concentrate into intermittent streams along the deepest parts of the slope. Confined to narrow gullies, such flows would unlikely erode comparable amounts of organic debris. Moreover, the sedimentological properties of the event layers are notably consistent throughout the record, indicating deposition by a single dominant process. This is reinforced by the PCA, which separates only two groups, background sedimentation and event layers, without evidence for a second type of event deposit, unlike what we observed using a similar approach in Hardeng et al. (2022).

For these reasons, we maintain that snow avalanches are the most plausible and consistent explanation for the event layers in Lake Vatnasetvatnet, while acknowledging that short-lived floods may contribute to increased influx of allochthonous sediments, but not enough to be recognised as a distinct event layer by our proxies.

In fact, the paper does not describe in detail the sedimentologic characteristics of the event layers. Fig. 5 is very nice, and displays for one case the detailed sediment structures, which is, however, in my view highly similar to many well-described flood turbidites from comparable lakes. There is the sharp base, a basal sand, a graded unit, disperse organic matter and a fining-upward cap (why is "inverse graded" on Fig. 5 and "fining upward" in the text?).

We respectfully disagree that the manuscript lacks a detailed description of the sedimentological characteristics of the event layers. The sedimentology and measured properties are described thoroughly in Section 4.2 (Core stratigraphy, lines 329–339), and further visualised in several figures, specifically, the data plots in Fig. 5, the PCA in Fig. 6, and the CT visualisation in Fig. 7. The link between these sedimentological characteristics and the process behind them is further discussed in section 5.2. We believe the reviewer may be referring to the CT rendering in Fig. 7 rather than Fig. 5, and we appreciate the positive assessment of this figure.

We agree that the event layers are similar to flood layers described in many published records. However, when viewed in the geomorphological and hydrological context of the catchment, the most consistent interpretation remains snow-avalanche deposition. The key argument for an avalanche origin lies in the catchment characteristics rather than in the

sedimentary structures themselves. The limited drainage area, compact till, and vegetated slopes strongly constrain fluvial sediment transport, whereas these same features favour frequent snow-avalanche activity and efficient sediment entrainment across the slopes. The interpretation of event layers in lacustrine records always depends on detailed knowledge of the depositional context and sediment sources. In our case, the morphology of the colluvial fan entering the lake closely matches avalanche-derived deposits described in previous studies (e.g., Rapp, 1960; Luckman, 1978; Blikra & Nemec, 1998), and the depositional style and sedimentology of the snow-avalanche layers correspond well with those reported by e.g., Sabatier et al. (2022).

Regarding the "inverse graded" label in Fig. 7 (not Fig. 5), we thank the reviewer for catching this oversight. It should read "graded" and will be corrected in a revised version.

Some high-density inversions with thin layers towards the base of the event deposit (beautifully shown on the CT data) may also indicate the waxing and waning of the flood (well described in literature), a process that I cannot imagine well in an avalanche. So I am very reluctant to assign all these layers to avalanches. As this is the key interpretation of this study, I expect at least a detailed discussion and explanation, showing more sedimentologic features of the layers, why they cannot be deposited by floods.

In this setting, a flood layer, if it existed, would not display the waxing and waning typically observed in larger catchments, where floods build toward a culmination before gradually diminishing. Here, the activation of the small intermittent streams would require extreme precipitation events/torrential rainfall, and the steep terrain and proximity to the lake would result in an almost instantaneous response, more comparable to a debris flow. In Norwegian, such events are referred to as 'flomskred' ("flood avalanches").

We argue against this process in the Vatnasetvatnet catchment, as there is no geomorphological or sedimentological evidence on land to support it. Such events would be far more channelised than snow avalanches, and the lowest parts of the gullies, where flowing water would concentrate, lack any significant sediment or soil cover. These areas were likely flushed out during deglaciation and subsequent paraglacial activity. In contrast, compact till and surface debris on the surrounding slopes provide sufficient material for mobilisation by snow avalanches, which have a much broader impact area than any potential flood or flood avalanche/debris-flow event.

There must be avalanche-specific features of event layers in the literature, but what I see seems highly reminiscent of flood layers. Several types of avalanches are mentioned (ice-free lake, on frozen lake, wet snow, rain-induced), is it also possible that they can be discerned? In the end, all layers are solely considered to be wet avalanches in an unfrozen lake and I am not convinced that this is the case. This uncertainty is also caused by the lack of an instrumental or historic calibration, i.e. a ground truthing that identifies a recent prominent layer and correlates it to a known event. Not sure such time series exist for Lake Vatnasetvatnet, but it should be attempted or at least commented why this is not possible.

While we cannot explicitly rule out the contribution of other processes (very few studies can), our combined lines of evidence point toward deposition by wet snow avalanches, either entering an ice-free lake or impacting and breaching lake ice. As noted in lines 475–479: "While the sharp basal boundary may indicate erosion, the age-depth model does not indicate significant hiatuses, suggesting that these contacts reflect compaction or settling

dynamics rather than reworking. Their irregular basal contacts argue against deposition as dropstones from melting lake ice (Luckman, 1975; Nesje et al., 2007). Instead, these features point to sudden, high-momentum sediment delivery by water-saturated snowflows that entered the lake directly or, potentially, by breaching lake ice upon impact (Sabatier et al., 2022)."

Dry snow slab and powder avalanches are less likely to produce comparable deposits, as they are less erosive and typically mobilise only limited amounts of sediment (e.g., Rapp, 1960; Blikra et al., 1998; Nesje et al., 2007). Although such avalanches have almost certainly occurred throughout the Holocene, they are less likely to form distinct event layers, as they tend to slide along glide planes within the snowpack rather than eroding vegetation and surface sediments on the slopes. The consistency in sedimentological characteristics among the event layers further supports deposition by a single dominant process. Considering all the described evidence, including the geomorphological setting, sedimentological characteristics, and the uniformity of event-layer properties, we interpret the layers as representing repeated wet-snow avalanche events.

Regarding instrumental or historical calibration, we fully agree that such data would be highly valuable. However, there are no documented avalanche events from this specific catchment that can be correlated to the identified layers. Nonetheless, we have observed and photographed modern snow avalanche accumulation on the colluvial fan and extending into the lake (Fig. 4D), which provides qualitative support for ongoing avalanche activity. We acknowledge that this image is of limited quality, but it remains the only photographic documentation available, supplemented by local observations confirming recurrent avalanche activity in recent decades. As reported in section 4.1, there are however multiple geomorphic and sedimentological evidence of recent snow avalanche activity.

What is also is intriguing in this context, is that the avalanche record does match closely the flood records of other Norwegian lakes (Fig. 9). In all three distinguished Holocene periods, the authors mentions matching records of snowmelt floods chronologies, so the pattern is the same. This leaves me uncertain whether avalanches really match flood records, or whether we indeed look rather at an avalanche or a combined record.

It is indeed intriguing that the reconstructed avalanche frequency corresponds closely with flood chronologies from other Norwegian records. We argue that this parallel reflects a common climatic driver. Both snow avalanches and snowmelt-dominated floods respond to winters characterised by heavy snowfall and frequent winter storms. Thus, a coherent pattern between the two proxies is expected under conditions of enhanced winter precipitation and storminess. To clarify, our record aligns with flood reconstructions from river systems primarily influenced by snowmelt, but not with those dominated by rainfall-induced floods. This distinction supports the interpretation that both proxies reflect variations in snow accumulation and winter hydroclimate rather than direct process overlap.

It is also important to note that any flood occurring within the Vatnasetvatnet catchment would necessarily be rainfall-induced. In a small, steep alpine basin like this, snowmelt typically drains gradually and diffusely across the slopes rather than concentrating into sustained channel flow. The catchment lacks both a substantial accumulation area and any capacity for prolonged meltwater storage, meaning that the hydraulic thresholds required to generate a flood capable of depositing a distinct sediment layer would not be reached.

As last general comment, I would say that I rather recognize four Holocene periods with characteristic event layer patterns: The 6500 to 4200 cal BP window is summarized in one epoch, but I do recognize a period 5500-4200 as an interval with intriguing lack of event layers, and 6500-5500 as an interval with very high activity (Fig. 9).

We thank the reviewer for this observation. We agree that the interval between 6500 and 4200 cal BP shows internal variability, including relatively low activity around 5300–4200 cal BP. However, we deliberately chose to describe three main Holocene periods to emphasise the broad, climatically coherent phases comparable across regional records.

We view the interval 6500–4200 cal BP as a single, transitional phase, shifting from the drycold winter conditions of the early Holocene to the more humid winters of the late Holocene. The instability within the 6500-4200 cal yr BP is indeed intriguing! And like we discussed in section 5.4, this instability likely reflects competing seasonal influences under orbital forcing: warm summers promoting snow- and ice-free slopes, while increasingly warm and humid winters enhanced avalanche potential. Therefore, the internal fluctuations within this period fit within the broader climatic regime and event frequency response seen in other studies during the middle Holocene. While a finer subdivision (e.g., 6500–5500 and 5500–4200 cal BP) could certainly be made, doing so risks overstating short-term variability that is not consistently expressed across regional records. We aimed to present three main phases that align with large-scale climatic transitions and provide a clear framework for comparison with other palaeoclimate and geohazard reconstructions in Norway.

**Some Detailed/technical comments**

55: The authors provide as examples of sediment-based avalanche reconstructions four Norwegian and one Alpine case studies. Considering the discussion above on distinction between flood and avalanche deposits, a summary of key criteria and key sedimentary structures would be very helpful here.

While several distinguishing features of flood and avalanche deposits have been described in the literature, no single diagnostic criterion can unambiguously determine the type of event responsible for a given layer. The characteristics of such layers are highly site-specific and depend strongly on local factors such as sediment sources, slope morphology, and lake-basin geometry. Nevertheless, a short summary of key sedimentological criteria could be useful, and we can provide a brief overview of the general tendencies reported from comparable study sites in a revised version.

173: Why is this discussion of the cirque (barely shown on Fig. 2) so important? This is not marked as avalanche track. It is interesting for the Late Glacial situation, ok, but if you mention it, the cirque should be shown fully on a slightly larger map.

We appreciate the reviewer's comment, but we are uncertain why the discussion of the cirque was perceived as particularly prominent or "important". It is not a central element of the study but was briefly mentioned because we mapped a moraine ridge near the lake that helps explain the final glacial input recorded as the transition from Unit B to Unit A in the sediment cores. The cirque is relevant mainly for constraining the timing of the final deglaciation and establishing that any glacial influence on sedimentation ceased when this cirque melted away in the early Holocene. Including this brief discussion also provides context for ruling out later glacial contributions to the sediment record.

Figure 5 is very well done, nice. I fully agree with the interpretation of the sharp lithologic transition coinciding with the disappearance of the glacier in the catchment, as supported by the age model. But I got a bit lost in the description of the stratigraphy. Four cores were retrieved, two were merged into one composite section (VABG) in the western basin (122, 109), this is fine. The shorter one was taken to obtain the sediment-water interface, but why was it not sampled at same location as the piston core? Then two cores were recovered from the eastern basin (209, 309), but only one is shown in Figure 5 (209) but 309 is discussed as well in the text. It is all a bit hard to follow.

We thank the reviewer for the positive assessment of Figure 5 and the stratigraphic interpretation. We note that the reviewer's summary in fact captures our description accurately, which makes it somewhat unclear what aspect was perceived as difficult to follow. Nevertheless, we appreciate the opportunity to clarify these points.

The shorter core (VAG122) was collected 13 years after the piston core (VAP109) to obtain an undisturbed sediment-water interface. The coring was conducted from lake ice, and we were unable to core the exact same location. However, the cores were retrieved in close proximity to each other, in the same basin at almost the same depth.

Core VAP309 is not included in the discussion. It is mentioned in the results (4.2) only to document its existence and to provide a rationale as to why it was excluded from further analysis; due to its proximity to the colluvial fan and higher likelihood of containing disturbed or reworked material. Instead, we relied on core VAP209 from a more distal and undisturbed position taken from the same (eastern) basin.

330: "and represent instantaneous deposits, i.e., event layers". This statement is here a bit premature. You describe here the sediment, and start right away with an interpretation of depositional environment but one needs to develop the case why these are event deposits. A more detailed description of sedimentary features will help.

We agree that the term "event layers" may appear somewhat interpretative at this stage of the results. Our intention, however, was not to imply a specific depositional process but to introduce a consistent term for the layers that are later quantified and discussed in detail. Because the subsequent results include quantitative identification and counting of these layers, we found it practical to use the term "event layers" throughout for clarity. If this wording is considered too interpretative, we can revise it to "distinct minerogenic layers" or similar.

Regarding the request for a more detailed description, we believe the manuscript already provides a comprehensive account of the sedimentary characteristics (lines 329–339). This includes all measured properties (XRF, CT, DBD, LOI, grain size, and sorting), as well as visual sedimentological features such as basal contacts, grading, organic content, and macrofossil occurrence.

363: "For CT thresholding, we classified depths with more than 40% minerogenic sediment as event layers": So the 40 % are a strong quantitative criteria, but it is not clear how this value was obtained by the CT data.

The decision to use 40% as the threshold for classifying a depth interval as an event layer was based on an iterative empirical calibration in which CT density contrasts were compared

with the visual core log and independent proxies (MS, DBD, XRF). This value provided the most consistent correspondence between the CT data and visually identified minerogenic layers. If the reviewer is referring to the technical procedure for quantifying the volume percentage of materials in the CT dataset, we can clarify this in the methods section by elaborating on how the greyscale thresholding and voxel classification were applied.

364: "Layers >3 mm thick were marked as instantaneous deposits (*slumps*) in the rBacon agedepth model to improve chronological accuracy and better reflect true depositional ages (Fig. 8)". The term "slump" is not an appropriate term here, as it a remobilized slope deposits. This is maybe a fix term in the Bacon software, but it should not be used here, these are not slumps.

Thank you for addressing this. We are aware that "slump" is not an appropriate sedimentological term in this context. It is merely a technical and widely used setting within the rBacon package in R, where "slump" refers to any instantaneous or rapidly deposited layer. We can clarify this to avoid confusion.

The GPR profile are not as clear as they are interpreted. It would be useful to make them larger: do we really see these 6 m of sediments? Maybe reflection seismic surveys would give better picture of the sediment architecture, but I guess the authors make the best out of their available site survey data. On Fig. 2a, the cores should be labelled as well. I wonder why were cores were not taken on GPR profiles? Consequently, the cores were obviously projected on the radar lines, this should mentioned.

It is unclear whether the reviewer refers to the readability of the figure or to the interpretation itself. If the concern relates to readability, we can separate Figures 2a and 2b and enlarge the GPR profiles for improved clarity. The GPR data were collected in 2009, and we indeed tried to make the best use of the available material. The cores are labelled in Figure 2a, but the reviewer may be referring to the smaller overview map in Figure 2b. We can include labels on this small map as well for clarity. The primary purpose of the GPR survey was to map lake bathymetry, as this was the only possible approach from lake ice. When selecting coring sites, factors such as water depth, basin morphology, and proximity to avalanche tracks were prioritised over coring directly on the GPR lines. We agree that core locations were projected onto the radar profiles; we can state this explicitly in the caption.

1-sigma is rather confident for the age model, usually, such lacustrine age models are done with a 2-sigma confidence interval. This just as a remark, in this case, it would not change the interpretation a lot, I guess.

The radiocarbon ages reported in Table 1, and the ages used in the age-depth model are the 2-sigma calibrated ranges, not 1-sigma as stated by the reviewer. In Table 1, the minimum and maximum ages represent the lower and upper bounds of the 2-sigma range (-1 sigma and +1 sigma, respectively), while the median probability is provided for reference.

In summary, I like the paper but would like to see the key issue addressed, i.e. the distinction between flood and avalanche deposits. Overall, I look forward to see this published after moderate (I only can chose minor or major, so I made minor) revisions.

We thank the reviewer for the constructive and encouraging feedback. We appreciate the positive overall assessment of the manuscript, and we value the reviewer's perspectives and

agree that the distinction between flood and avalanche deposits is an important aspect to clarify. As outlined in our responses above, we are confident that the event layers reflect snow avalanche activity, based on the combined geomorphological and sedimentological evidence. However, we are prepared to make targeted clarifications and adjustments in the text to ensure that this reasoning is more explicitly conveyed.

---

## Author Comment (AC3)

**Anonymous Referee #2:**

(Referee comments in regular font, author response in italic gray font)

I thoroughly enjoyed reading the manuscript entitled "Decoupling climate and avalanche activity: Holocene insights from lacustrine sediments in western Norway". The manuscript is well-written and presents multi-proxy reconstruction of Holocene snow avalanche activity using lacustrine sediments from Lake Vatnasetvatnet in western Norway. The study employs sedimentological, geochemical, and CT scanning techniques to reconstruct ~10,000 years of snow avalanche activity from lake sediment records, which seems methodologically sound. Avalanche event layers are identified using two semi-automated approaches: (1) rate of change in Ti concentrations and (2) thresholding of CT grayscale data. Based on these methods, the authors distinguish three main phases of Holocene avalanche activity: (1) low activity during Early Holocene (>6500 cal yr BP), (2) increased activity during the mid-Holocene (6500–4200 cal yr BP), and (3) the highest frequency during Late Holocene (~4200 cal yr BP to the present). These patterns are interpreted as being primarily influenced by large-scale atmospheric circulation (i.e., variability in North Atlantic sea surface temperatures and fluctuations in the North Atlantic Oscillation). This study represents a valuable contribution to palaeoclimatology and natural hazard research, particularly in a region where long-term avalanche dynamics are poorly constrained.

**Strengths**

The manuscript demonstrates several notable strengths: (1) the 10,000-year record are particularly impressive in avalanche research, (2) combined use of LOI, DBD, grain-size analysis with magnetic susceptibility, XRF, and CT scanning greatly enhances the robustness of event layer identification, (3) the application of dual detection methods (Ti-based rate of change and CT thresholding), helps to minimize subjectivity in distinguishing event layers, (4) the chronostratigraphy is supported by AMS dating and rBacon modeling, and (4) the comparison of avalanche frequency with regional climate systems provides valuable context and strengthens the broader climatic interpretation of the findings.

We sincerely thank reviewer 2 for the positive and encouraging assessment of our manuscript. We appreciate the recognition of the study's methodological approach, robustness, and contribution to improving the understanding of long-term avalanche dynamics in western Norway. We are particularly pleased that the reviewer highlights the value of the multi-proxy design, the detection methods for identifying event layers, and the integration of the avalanche record within a broader climatic framework.

**Limitations/improvement**

Despite these strengths, several areas of the manuscript could benefit from further clarification and refinement: (1) The use of the term decoupling in the title seems not fully explored. The manuscript primarily demonstrates co-variability between avalanche activity and climate proxies rather than a clear decoupling. (2) More explicit discussion of uncertainties, especially in threshold calibration and sediment source attribution would be nice. (3) The novelty as highlighted in the manuscript may seem overstated. Similar reconstructions studies exist, and this study can be discussed as more on refinement than on conceptual breakthroughs.

- 1) We appreciate this observation and understand the concern. In this context, decoupling refers to a weakening or alteration of an expected relationship between two variables that are normally linked, in this case, between climate variability and avalanche frequency. It highlights that avalanche activity does not always respond linearly or synchronously to temperature-driven climate forcing. While the manuscript documents several intervals of covariability between avalanche frequency and large-scale climate modes (e.g., the NAO), it also identifies periods where this relationship appears weakened or non-linear, reflecting shifts in the relative influence of temperature and precipitation. We are open to considering alternative, more precise titles should the editor find this advisable. Other suggestions: "10,000 years of snow avalanche activity in Western Norway: A multi-proxy lake sediment record from lake Vatnasetevatn, Hardanger". Or alternatively: "North Atlantic climate variability and Holocene snow avalanche activity in Western Norway recorded in lake sediments".
- 2) We agree that a more explicit discussion of uncertainties would strengthen the manuscript. The CT- and Ti-based thresholds were calibrated empirically by comparing the automated detection outputs with visually identified layers, as described in lines 495–496 and 501–506. We can elaborate on the uncertainties in a revised version. Regarding sediment source attribution, we refer to our responses to Reviewer 1 for a detailed rationale; in short, the geomorphological setting strongly indicates snow avalanches as the dominant process, although minor contributions from fluvial reworking cannot be entirely excluded.
- 3) It is unclear which specific formulation(s) reviewer 2 is referring to here. We respectfully disagree that the manuscript overstates its novelty, we do not claim any conceptual breakthrough, nor do we highlight novelty beyond the context of applying established multiproxy techniques to a previously unstudied catchment. The manuscript explicitly situates our work within the framework of earlier avalanche-related sediment studies in Norway and elsewhere.

For instance, lines 54–58 clearly acknowledge previous snow avalanche reconstructions from western Norway (e.g., Blikra & Nemec 1998; Nesje et al. 2007; Vasskog et al. 2011; Aa et al. 2022), and from the Alps (e.g., Fouinat et al. 2017). Likewise, our methodological discussion (489–503) directly references earlier studies that have employed RoC analysis (Støren et al., 2010; Røthe et al., 2019b; Johansson et al., 2020; Hardeng et al., 2022), and CT scanning (van der Bilt et al., 2021; Cederstrøm et al., 2021; Ballo et al., 2023; Støren et al., 2010; Hardeng et al., 2022; 2024; Fouinat et al., 2017; Røthe et al., 2019b) for detecting sedimentological structures and event deposits.

**Other Comments**

All event layers are interpreted as avalanche deposits, but what is the likelihood that some of these layers were instead formed by fluvial processes, erosion, or landslides coupled with fluvial activity? While the surrounding landscape may not support a large fluvial system, smaller streams, in combination with upstream erosion and landsliding during climatic extremes, could plausibly produce similar deposits. Would more detailed analyses of the sedimentary characteristics help to distinguish between these potential depositional processes? For example - It was discussed that >3m layers are related to the slumps - are those sudden deposits derived from slumps upstream? Clarification on why these layers are slump-related would be helpful. Given that, I felt some discussion on alternative sediment

sources, such as debris flows, rockfalls, fluvial pulses as well as vegetation dynamics and anthropogenic influences (i.e. during the late Holocene) would be appreciated.

We thank the reviewer for this comment, which aligns with similar points raised by Reviewer 1. We acknowledge that our rationale for excluding fluvial or landslide-related processes could be presented more clearly, and we are happy to expand this discussion in the revised manuscript using the arguments outlined in our detailed response to Reviewer 1. Extensive geomorphological mapping shows no evidence of landslides, debris flows, or fluvial erosion within the catchment.

Reviewer 2 states that "It was discussed that >3 m layers are related to the slumps...", this is not correct. The manuscript reads (Line 364-366): "Layers >3 mm thick were marked as instantaneous deposits (slumps) in the rBacon age-depth model to improve chronological accuracy and better reflect true depositional ages (Fig. 8)." We are never discussing 3 m thick slumps; we are referring to the thin (3 mm-10 cm) event layers that we classified as "slumps" within the rBacon age-depth modelling software. We acknowledge that this terminology could cause confusion and acknowledge that this should be clarified.

We emphasise that the described event layers are not slump-related deposits. The word "slump" is only mentioned once in the manuscript, and that is in relation to settings within the rBacon age-depth modelling. Our interpretation is that the layers are deposited by snow avalanches triggered in the upper slopes (Avalanceh tracks 1 and 2, Fig. 2, 3, and 4). We describe the depositional process and supporting sedimentological evidence in depth in lines 466–479.

The role of solar forcing is discussed but not quantitively modeled. A more rigorous climate-forcing attribution or discussion would strengthen the manuscript.

We appreciate the reviewer's suggestion. We agree that a quantitative climate-forcing attribution would further strengthen the mechanistic understanding of the observed variability. However, such modelling is beyond the scope and aims of this study, which focuses on the sedimentary reconstruction and its interpretation in relation to established palaeoclimate records. Our discussion of solar forcing is therefore intentionally qualitative, based on well-documented orbital and insolation trends (e.g., Laskar et al., 2004) and their established influence on Holocene hydroclimate variability in the North Atlantic region. We believe this contextual approach is appropriate for the present dataset and ensures the manuscript remains focused on the reconstruction and interpretation of avalanche activity rather than extending into climate modelling or quantitative attribution analyses, which would require a different methodological framework and data resolution.

Ti as a proxy for minerogenic input is well-established, but its specificity to avalanche deposits may require further discussion. I assume, Ti can also reflect fluvial or colluvial processes too. Similarly, the selection of thresholds (i.e., 95th percentile for Ti RoC and the 40% CT thresholding) lacks sufficient justification. A sensitivity analysis or probabilistic modeling approach could help validate these thresholds and reduce subjectivity.

We agree that Ti is a general indicator of minerogenic input and not inherently specific to avalanche deposits. In our study, Ti was not used to distinguish between depositional processes per se but to objectively identify minerogenic layers within the sediment sequence. The process attribution, whether these layers represent snow avalanches or alternative mechanisms, is discussed separately, based on the geomorphological and sedimentological context of the catchment.

We do not agree that the threshold selection lacks justification. Both thresholds (the 95th percentile for Ti rate-of-change and the 40% CT threshold) were calibrated through iterative comparison with visually identified minerogenic layers and supporting sediment proxies (e.g., MS, DBD, LOI), yielding consistent results between two independent methods (r = 0.76). This strong agreement demonstrates the robustness of our approach and aligns with procedures applied in comparable studies. While threshold determination inevitably involves some subjectivity, a probabilistic or sensitivity-based calibration would require independent "ground truth" data unavailable for this type of archive. Our approach provides a transparent and reproducible framework for event-layer identification within these constraints.

Although the manuscript discusses different avalanche types (e.g., wet vs. dry), no attempt is made to differentiate them within the sedimentary record. If feasible, distinguishing between avalanche types based on sediment characteristics would add valuable depth to the interpretation.

We respectfully note that attempts to distinguish between different depositional processes were indeed made through the Principal Component Analysis (PCA), which is designed to detect compositional variability within the dataset. The PCA successfully identified three sediment types, glacially derived silt (Unit B), background sedimentation, and event layers (Unit A). The event layers share the same sedimentological characteristics, indicating deposition by a single dominant process. This uniformity, supported by both visual inspection and statistical analysis, suggests that further subdivision into separate avalanche types (e.g., wet vs. dry) is not warranted based on the available sedimentary evidence.

The manuscript should clarify how slope gradients were calculated - if derived from a digital elevation model (DEM), the type and horizontal resolution should be specified. The identification of two avalanche tracks is well-supported, but the possibility of lateral migration or undocumented tracks should be acknowledged.

The slope gradients were derived from the LiDAR-based digital elevation model (0.25 m resolution) provided by the Norwegian Mapping Authority, as stated in lines 113–115. We can clarify this further in a revised version by explicitly describing how slope gradients were calculated from the DEM.

Regarding the possibility of lateral migration or undocumented avalanche tracks, we consider this unlikely. The steep topography of the southern catchment strongly constrains avalanche paths, and the treeless corridors marking Tracks 1 and 2 correspond clearly with geomorphic and sedimentological evidence of recurrent activity. No additional tracks were observed in the high-resolution LiDAR data or during field mapping, although we acknowledge that minor variations in runout width and extent likely occur between individual events.

Fig. 1C - The horizontal blue lines likely stem from ESRI data artifacts. I would suggest regenerating the map using better DEM data. Also, the catchment boundary should be clearly labeled and described.

The horizontal blue lines in Fig. 1C are not ESRI data artifacts but represent mapped bogs and marshes derived from national land-cover data. We agree that this could be

misinterpreted in the absence of a legend and will clarify or remove these symbols in a revised figure, as they are not essential to the discussion.

The DEM used for this map is a LiDAR-derived dataset with 0.25 m resolution from the Norwegian Mapping Authority (www.hoydedata.no), which represents the highest-quality topographic data available for this region. Regarding the catchment boundary, it is already shown as the black outline around Lake Vatnasetvatnet and described in the figure caption, but we can explicitly label it in the revised version if necessary to avoid confusion.

**Fig. 2B (GPR Profiles): if possible, labeling the water-sediment and sediment-bedrock interfaces would help readers assess the effectiveness of the GPR method.**

We thank the reviewer for this helpful suggestion. We agree that labeling the water-sediment and sediment-bedrock interfaces will improve the clarity of Fig. 2B.

**Fig. 5 –26th layer appears missing.**

We thank the reviewer for this careful observation. The reference to the 26th layer in the caption is a typographical error, it should read 25, consistent with the sediment log and core description provided in Section 4.2. This will be corrected in the revised version.

**Fig. 7: More visual clarity would be helpful, for example through clear labeling of the cores.**

We thank the reviewer for this comment. Figure 7 is a CT visualisation of the interval 90–110 cm in core VAPG, as stated in the caption. To improve clarity, we can add "VAPG" directly to the figure itself. It is somewhat unclear what other aspects of the figure the reviewer found visually unclear, but we will review the layout and contrast to ensure that key features are as legible as possible in the revised version.

To further enhance the manuscript, I recommend incorporating additional sedimentological and geochemical indicators to better distinguish avalanche deposits from other high-energy events. A probabilistic framework for threshold selection and age-depth modeling could improve the robustness of the reconstruction. Integrating climate modeling and linking avalanche frequency to regional climate simulations would provide valuable insights into future scenario analyses. Finally, incorporating land-use history and vegetation dynamics more explicitly would help contextualize the Late Holocene trends. In conclusion, this is a well-written, well-structured and timely study with significant potential to advance our understanding of long-term avalanche dynamics. However, I think the interpretation linking depositional layers exclusively to avalanche events requires more sedimentological justification and differentiation from other depositional processes. I believe addressing the points outlined above would greatly enhance the clarity, robustness, and impact of the manuscript.

We thank the reviewer for their constructive comments, and we appreciate the recognition of the study's overall quality and potential contribution. We acknowledge that additional sedimentological and geochemical analyses (e.g., mineralogical or isotopic indicators) could provide further insights into the depositional mechanisms. However, incorporating such datasets lies beyond the scope of the present study, which was designed to establish a long-term, first-order reconstruction of snow-avalanche activity. The integration of probabilistic or climate-modeling frameworks, as well as detailed land-use reconstruction, would certainly

add further depth, but would require data types and resolutions not available for this catchment. We regard these as valuable directions for future research.

As detailed in our responses above, we are prepared to strengthen the discussion on alternative depositional processes and clarify the geomorphological and sedimentological reasoning behind our interpretation of the event layers as avalanche-derived. Together, these clarifications should improve both the transparency and robustness of the interpretation while keeping the study focused within its intended scope.